# *Dalbergia ecastaphyllum* (L.) Taub. and *Symphonia globulifera* L.f.: The Botanical Sources of Isoflavonoids and Benzophenones in Brazilian Red Propolis

**DOI:** 10.3390/molecules25092060

**Published:** 2020-04-28

**Authors:** Gari Vidal Ccana-Ccapatinta, Jennyfer Andrea Aldana Mejía, Matheus Hikaru Tanimoto, Milton Groppo, Jean Carlos Andrade Sarmento de Carvalho, Jairo Kenupp Bastos

**Affiliations:** 1Laboratory of Pharmacognosy, School of Pharmaceutical Sciences of Ribeirão Preto, University of São Paulo (USP), Av. do Café s/n, Ribeirão Preto 14040-903, SP, Brazil; 2Laboratory of Plant Systematics, Department of Biology, Faculty of Philosophy, Sciences and Letters at Ribeirão Preto, USP, Av. dos Bandeirantes 3900, Ribeirão Preto 14040-901, SP, Brazil; 3Cooperativa de Apicultores de Canavieiras (COAPER), Av. Burundanga 1900, Canavieiras 45860-000, BA, Brazil

**Keywords:** isoflavonoids, polyisoprenylated benzophenones, propolis, botanical sources

## Abstract

The Brazilian red propolis (BRP) constitutes an important commercial asset for northeast Brazilian beekeepers. The role of *Dalbergia ecastaphyllum* (L.) Taub. (Fabaceae) as the main botanical source of this propolis has been previously confirmed. However, in addition to isoflavonoids and other phenolics, which are present in the resin of *D. ecastaphyllum*, samples of BRP are reported to contain substantial amounts of polyprenylated benzophenones, whose botanical source was unknown. Therefore, field surveys, phytochemical and chromatographic analyses were undertaken to confirm the botanical sources of the red propolis produced in apiaries located in Canavieiras, Bahia, Brazil. The results confirmed *D. ecastaphyllum* as the botanical source of liquiritigenin (**1**), isoliquiritigenin (**2**), formononetin (**3**), vestitol (**4**), neovestitol (**5**), medicarpin (**6**), and 7-*O*-neovestitol (**7**), while *Symphonia globulifera* L.f. (Clusiaceae) is herein reported for the first time as the botanical source of polyprenylated benzophenones, mainly guttiferone E (**8**) and oblongifolin B (**9**), as well as the triterpenoids *β*-amyrin (**10**) and glutinol (**11**). The chemotaxonomic and economic significance of the occurrence of polyprenylated benzophenones in red propolis is discussed.

## 1. Introduction

The red propolis is the second most produced and traded type of propolis in Brazil and constitutes an important commercial asset for northeast Brazilian beekeepers. Apiaries devoted to Brazilian red propolis (BRP) production are frequently located around native populations of the fabaceous species *Dalbergia ecastaphyllum* (L.) Taub., which produces a red resin that is collected by bees to produce propolis in the beehive [1]. The studies by Daugsch et al. (2008) and Silva et al. (2008) were the first ones to describe, simultaneously in 2008, *D. ecastaphyllum* as the main botanical source of the BRP [2,3]. These initial studies together with the report of Piccinelli et al. (2011) confirmed the presence of a rich variety of phenolic compounds, in both the propolis and the plant resin, such as chalcones (e.g., isoliquiritigenin), flavonoids (e.g., luteolin, liquiritigenin), isoflavones (e.g., formononetin, biochanin A), isoflavans (e.g., vestitol, neovestitol, 7-O-methylvestitol), pterocarpanes (e.g., medicarpin, homopterocarpin, vesticarpan), and C30 isoflavans (retusapurpurins A and B) [4]. 

Subsequent studies have shown the great qualitative and quantitative variability of the constituents present in red propolis, which are influenced by regional and seasonal factors [5]. Nevertheless, reports on the presence of substantial amounts of polyprenylated benzophenones, such as gutifferone E and oblongifolin A in BRP, have appeared in the literature more frequently [6,7]. Even though the botanical source of these latter constituents was unknown, Piccinelli et al. (2011) inferred from a chemotaxonomic point of view that these compounds must derive from a member of the Clusiaceae family [4]. Important biological activities of BRP extracts, such as antimicrobial, fungicidal, and cytotoxic properties, have been correlated with the occurrence of polyprenylated benzophenones [6,8,9]. Thus, establishing its botanical source may help to increase red propolis production and to attain a higher degree of chemical standardization. Therefore, field surveys, phytochemical and chromatographic analyses were undertaken in order to identify the botanical source of polyprenylated benzophenones present in the red propolis produced by apiaries of the Beekeepers Association of Canavieiras (Cooperativa de Apicultores de Canavieiras—COAPER), Bahia, Brazil.

## 2. Results

A filed survey was carried out in March 2019 in order to collect local red propolis samples, the reddish resin of *D. ecastaphyllum*, and to identify other resin-producing plant species, especially of the Clusiaceae family, located in the surrounding flora of apiaries from the COAPER beekeepers association. The presence of individuals of *Symphonia globulifera* L.f. was evidenced and its resin was collected, transported to the laboratory, lyophilized, and kept under refrigeration.

### 2.1. Isolation of Polyprenylated Benzophenones from Symphonia globulifera and BRP

Samples of both BRP and *S. globulifera* resin were extracted with aqueous ethanol 70% and submitted to partition with solvents of increasing polarities. The hexane fractions were separately submitted to chromatographic procedures that resulted in the isolation of two polyprenylated benzophenones (**8** and **9**) and two pentacyclic triterpenoids (**10** and **11**) from both fractions. The structures of the isolated compounds were established based on 1D and 2D-NMR spectroscopy, high-resolution MS, and comparison with the literature data for the polyprenylated benzophenones guttiferone E (**8**) [10], in a mixture with its double bond position isomer xanthochymol [11], oblongifolin B (**9**) [12], and the triterpenoids *β*-amyrin (**10**) [13] and glutinol (**11**) [14]. The NMR spectra of **8** and **9** in CDCl_3_ displayed abnormalities that disappeared when spectra were recorded in CD_3_OD + 0.1% trifluoroacetic acid (TFA). The chemical structure of those isolated constituents (**8**–**11**), as well as other phenolic compounds (**1**–**7**) identified in BRP are displayed in Figure 1.

The oblongifolins were first isolated from *Garcinia oblongifolia* by Hamed et al. (2006) [12]. The occurrence of oblongifolin A in samples of BRP was initially reported by Piccinelli et al. (2011), while further publications replicated this finding by LC-MS analyses. However, mass spectrometry of oblongifolins is not enough to identify them, since they bear the same molecular weight, while none of those previous reports have provided NMR data to confirm the occurrence of oblongifolin A in BRP [4,6,7,8]. Therefore, the NMR spectroscopic data of the isolated compound **9** in comparison with published data of oblongifolins A and B is presented in Appendix A. The ^1^H and ^13^C NMR chemical shifts of these two stereoisomers are almost identical. Nonetheless, significant differences are found in the ^13^C NMR spectra of these two oblongifolins at δ_C-6_ (47.8 for A and 44.2 for B), δ_C-7_ (40.8 for A and 43.3 for B), and δ_C-8_ (61.8 for A and 64.3 for B). Similarly, the ^1^H NMR spectra of oblongifolin B displays a characteristic axial coupling constant for H-7 of 13.0 Hz. Therefore, the isolated compound **9** corresponds to oblongifolin B and its isolation from BRP is here reported for the first time.

### 2.2. The Botanical Sources of Brazilian Red Propolis

The chromatographic profiles of a typical red propolis sample and the resins of *D. ecastophyllum* and *S. globulifera* were recorded by a HPLC analysis method (Figure 2); the chromatographic parameters are described in the material and methods section.

Comparison of both the HPLC chromatographic retention times and its corresponding UV spectra with the ones of the standard compounds **1**–**9** were used for assigning the identity of peaks in the chromatographic profiles of the analyzed samples. The chromatographic prolife of a typical sample of BRP (Figure 2A) displayed at least nine main chromatographic peaks corresponding to phenolic compounds (**1** and **2**), isoflavonoids (**3**–**5**, **7**), a pterocarpane (**6**) and benzophenones (**8**–**9**). The resin of *D. ecastaphyllum* (Figure 2B) displayed the presence of chromatographic peaks mainly in the region of the HPLC chromatogram from 10 to 40 min, confirming *D. ecastaphyllum* as the botanical source of liquiritigenin (**1**), isoliquiritigenin (**2**), formononetin (**3**), vestitol (**4**), neovestitol (**5**), medicarpin (**6**), and 7-*O*-neovestitol (**7**). In the lipophilic region of the chromatogram (Figure 2C), the resin of *S. globulifera* displayed two main chromatographic peaks, which were assigned to guttiferone E (**8**) and oblongifolin B (**9**), confirming *S. globulifera* as the botanical source of polyprenylated benzophenones.

## 3. Discussion

The occurrence of the flavonoid liquiritigenin (**1**); the chalcone isoliquiritigenin (**2**); the isoflavone formononetin (**3**); the isoflavans vestitol (**4**), neovestitol (**5**), and 7-*O*-methylvestitol (**7**); as well as the pterocarpane medicarpin (**6**) have been previously reported both in BRP and in the resin of its botanical source, *D. ecastaphyllum* [4]. From a chemotaxonomic point of view, these constituents, especially the isoflavonoids, are characteristic chemotaxonomic markers for Papilionoideae (Fabaceae) subfamily members, to which the *Dalbergia* genus belongs [15]. The results reported here confirm that the main botanical source of the red propolis produced in the COPAER beekeepers association of Canavieiras is *D. ecastaphyllum*. This species, which occurs in coastal sand dune and mangrove ecosystems, is present in large populations in the surrounding areas of the apiaries not only in Bahia state but also in the northeast Brazilian localities where red propolis is produced.

On the other hand, the presence of the prenylated benzophenones guttiferone E (**8**) and oblongifolin A in commercial samples of BRP was previously reported [6,7,8] and proposed as a characteristic that could differentiate Brazilian samples from other red propolis, e.g., Cuban propolis [4]. However, the botanical source of polyprenylated benzophenones present in BRP samples was unknown, but it was inferred from a chemotaxonomic point of view that these compounds would be collected by bees from a resin-producing plant belonging to the Clusiaceae family [4]. The isolation of the polyprenylated benzophenones guttiferone E (**8**) and oblongifolin B (**9**) reported here from samples of BRP and the resin of *S. globulifera* confirms that the botanical source of these compounds is a member of the Clusiaceae family, in agreement with the chemotaxonomical inference of Piccinelli et al. (2011). The occurrence of compounds **8** and **9** in both BRP and *S. globulifera* was also confirmed here by HPLC analysis. Therefore, the present report identifies, for the first time, *S. globulifera* as the botanical source of polyprenylated benzophenones of BRP produced in the locality of Canavieiras, Bahia. Unlike the data previously reported, describing the occurrence of oblongifolin A in BRP, we identified, instead, its stereoisomer oblongifolin B (**9**), the occurrence of which in BRP and *S. globulifera* is here reported for the first time. Although *S. globulifera* is a well-known source of polyprenylated benzophenones, this is the first report on the occurrence of guttiferone E (**8**) in this species. Additionally, this report demonstrates the value of chemotaxonomy for establishing the botanical sources of propolis.

The knowledge of propolis plant sources can help to increase propolis production and to attain a higher degree of chemical standardization [16]. Unlike the Brazilian green propolis, the most produced propolis in Brazil, the market value of red propolis raw material is double the price of green propolis. Therefore, commercial and academic interest has been awakened to extend the area of bee pasture to increase the production of red propolis. The findings reported here demonstrate that BRP does not have only a single botanical source but at least two, *D. ecastaphyllum* and *S. globulifera*, which contribute different chemical constituents, principally isoflavonoids and polyprenylated benzophenones, which should be taken into account when installing new beehives for red propolis production in the northeast of Brazil.

## 4. Material and Methods

### 4.1. Field Survey

A field survey was carried out from 24 March to 5 April 2019 aiming to collect local red propolis samples, the resin of *D. ecastaphyllum*, and to identify other resin-producing species, especially of the Clusiaceae family, located in the surrounding flora of apiaries from the beekeepers association of Canavieiras (*Cooperativa de Apicultores de Canavieiras*—COAPER), Bahia, Brazil. The presence of individuals of *Symphonia globulifera* was evidenced, and their resins were collected, transported to the laboratory, lyophilized, and kept under refrigeration. The plant vouchers were identified and deposited as *Symphonia globulifera* L.f., SPFR 17770; *Dalbergia ecastaphyllum* (L.) Taub., SPFR 17,771 at the Herbarium SPFR of the Department of Biology, Faculty of Philosophy, Sciences and Letters at Ribeirão Preto, FFCLRP, University of São Paulo, USP.

### 4.2. Extraction and Isolation of Triterpenoids and Polyprenylated Benzophenones

Two hundred grams of BRP were extracted with aqueous ethanol 70% (Vetec Química, Rio de Janeiro, Brazil) for three times, 1:10 sample/solvent ratio, and g/mL furnishing 140 g of crude extract. The extract was then mixed with 200 g of microcrystalline cellulose (Sigma Aldrich, St. Louis, MO, USA) and submitted to solid-liquid partition with hexanes (Vetec Química, Rio de Janeiro, Brazil), three times of 500 mL, furnishing 23.8 g of hexane crude fraction. The hexane fraction was then submitted to vacuum liquid chromatography (VLC) by using 150 g of silica gel (Sigma Aldrich, St. Louis, MO, USA), 40–63 µm particle size, as the stationary phase and mixtures of increasing polarity of hexanes:ethyl acetate (100:0→30:70) as the mobile phase. The eluted fractions were monitored by thin layer chromatography (TLC) and pooled by similarity, generating four fractions: F1 (6 g), F2 (9.26 g), F3 (2.27), and F4 (1.1 g). TLC was carried out on precoated glass TLC silica gel 60F_254_ plates (Merck, Darmstadt, Germany), with detection accomplished by visualization with a UV lamp at 254 and 360 nm, followed by spraying with a 1% solution of 2-aminoethyl diphenylborinate in methanol (w/v). A portion of fraction F2 (500 mg) was submitted to centrifugal thin-layer chromatography on a Chromatotron device (Harrison Research, USA) by using a 1-mm disk of silica gel (10–40 µm particle size) as the stationary phase, and mixtures of increasing polarity of hexanes:ethyl acetate (100:0→30:70) as the mobile phase, affording compounds **8** (70 mg) and **9** (30 mg). A portion of fraction F1 (500 mg) was submitted to flash chromatography on an Isolera One equipment (Biotage, Sweden) by using a 10-g cartridge of silica gel (40 µm particle size) and mixtures of increasing polarity of hexanes:ethyl acetate (100:0→50:50) as the mobile phase, affording compounds **10** (30 mg) and **11** (40 mg). A portion of the *S. globulifera* resin (100 g) was also submitted to the procedures as describe above furnishing the same compounds **8**–**11**.

One-dimensional and 2-D NMR spectra of compounds **8**–**11** were acquired in a Brucker DRX500 NMR spectrometer (Brucker, Santa Barbara, CA, USA) operating at a frequency of 500 MHz for ^1^H and 125 MHz for ^13^C by using CDCl_3_ and CD_3_OD + 0.1% TFA as deuterated solvents (Sigma Aldrich, St. Louis, MO, USA). High-resolution mass spectrometry data of compounds **8** and **9** were obtained by direct infusion in negative ionization mode on an orbitrap mass spectrometer (Thermo Scientific, San Jose, CA, USA).

*Guttiferone E* (**8**): yellow amorphous powder; UV (HPLC-online) λ*_max_*: 249.6, 354.4 nm; ^1^H and ^13^C NMR data as reported by Gustafson et al., 1992 [10]. HRESIMS negative mode *m*/*z* 601.3546 [M - H]^−^ (calcd. for C_38_H_50_O_6_, 601.3529), 525.3232, 183.0116, 109.0284.

*Oblongifolin B* (**9**): yellow amorphous powder; UV (HPLC-online) λ*_max_*: 244.9, 350.9 nm; ^1^H and ^13^C NMR data as reported by Hamed et al., 2006 [12]. HRESIMS negative mode *m*/*z* 601.3546 [M - H]^−^ (calcd. for C_38_H_50_O_6_, 601.3529), 525.3232, 333.1349, 183.0116, 109.0284.

*β-Amyrin* (**10**): white needles; ^1^H and ^13^C NMR data as reported by Lima et al., 2004 [13].

*Glutinol* (**11**): white needles; ^1^H and ^13^C NMR data as reported by Mahato et al., 1981 [14]

### 4.3. HPLC Analyses of Botanical Sources and BRP

The chromatographic profiles of BRP samples, and the resins of *D. ecastaphyllum* and *S. globulifera* were obtained by using an HPLC-DAD method on a Waters 1500-series setup (Milford, CT, USA) with an Ascentis Express C18 column (2.7 μm, 150 × 4.60 mm) as the stationary phase and mixtures of water with 0.1% formic acid (B) and acetonitrile (B) as the mobile phase. The flow rate was set at 1.0 mL/min in a gradient elution mode as follows: 20→50% B in 40 min, 50→100% B in 90 min, 100% B (isocratic) until 95 min, 100→20% B until 100 min, 20% B (isocratic) up to 105 min. Benzophenone (20 µg/mL) was used as the internal standard (i.s.). One milligram of each resin sample or propolis extract was dissolved in methanol and subjected to chromatographic analysis. Standard compounds liquiritigenin (**1**), isoliquiritigenin (**2**), formononetin (**3**), vestitol (**4**), neovestitol (**5**), medicarpin (**6**), and 7-O-neovestitol (**7**) were available in the isolated compound library of the laboratory. The standard compounds **1**–**7** and the isolated compounds guttiferone E (**8**) and oblongifolin B (**9**) were dissolved in methanol (20 µg/mL) for HPLC analyses.

## Figures and Tables

**Figure 1 molecules-25-02060-f001:**
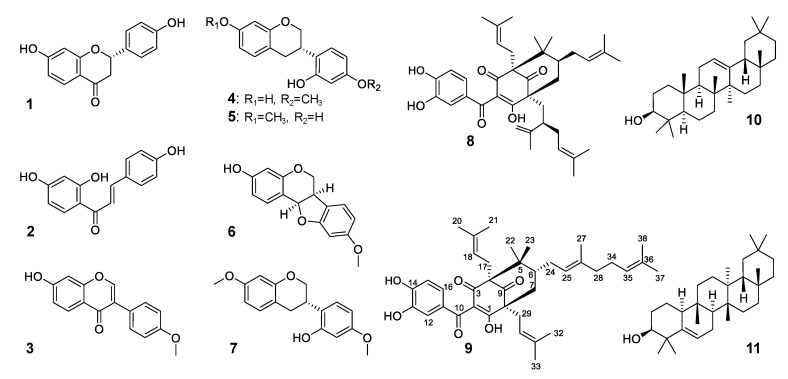
Chemical constituents of Brazilian red propolis.

**Figure 2 molecules-25-02060-f002:**
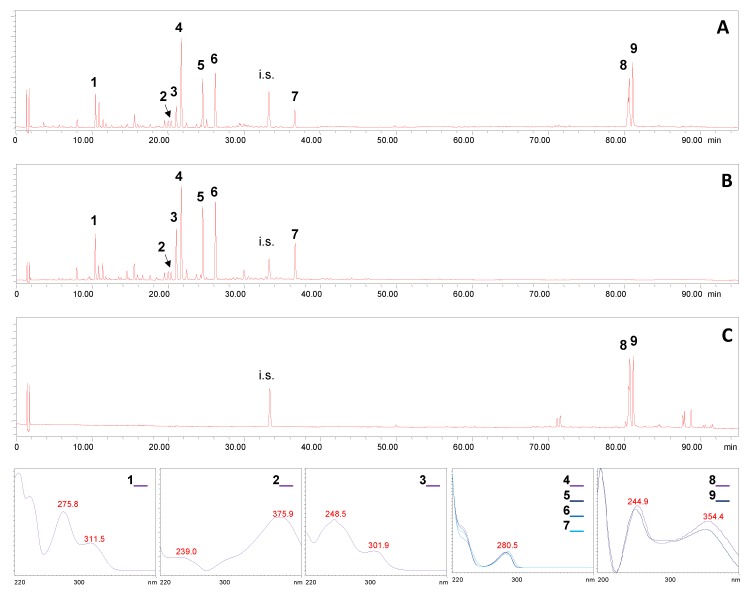
HPLC chromatographic profiles (275 nm) of Brazilian red propolis (**A**) in comparison with the resins of *Dalbergia ecastaphyllum* (**B**) and *Symphonia globulifera* (**C**). Numbers correspond to liquiritigenin (**1**), isoliquiritigenin (**2**), formononetin (**3**), vestitol (**4**), neovestitol (**5**), medicarpin (**6**), 7-*O*-neovestitol (**7**), guttiferone E (**8**), and oblongifolin B (**9**). UV spectra of compounds **1**–**9** are displayed at the bottom of the figure.

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
