# Peer review of "Dalbergia ecastaphyllum (L.) Taub. and Symphonia globulifera L.f.: The Botanical Sources of Isoflavonoids and Benzophenones in Brazilian Red Propolis"

_molecules, 2020, doi:10.3390/molecules25092060_

Round 1

Reviewer 1 Report

The present manuscripts report the study of two different Brazilian red propolis one from Dalbergia ecastaphyllum (L.) Taub. and  the other from Symphonia 3 globulifera L.f.. The authors conduct an identification flavonoid and benzophenones in the samples by HPLC analysis combined with HNM chemical characterization. In my opinion the paper is suitable for publication after minor revision.

The authors in section 2.2 says that "the structures of the isolated compounds were established based on NMR spectroscopy, high-resolution MS... but no HRMS data were report. Please specify how the HR-MS has been used and add this data (accurate mass, and fragmentation) to describe the isolated compounds.

Did the authors coupled the UV detector with mass spectrometry analysis? Please clarify this point

Author Response

Reviewer 1

COMMENT: The present manuscripts report the study of two different Brazilian red propolis one from Dalbergia ecastaphyllum (L.) Taub. and the other from Symphonia globulifera L.f. The authors conduct an identification flavonoid and benzophenones in the samples by HPLC analysis combined with HNM chemical characterization. In my opinion the paper is suitable for publication after minor revision.

ANSWER: We acknowledge the comments made by the reviewer.

COMMENT: The authors in section 2.2 says that "the structures of the isolated compounds were established based on NMR spectroscopy, high-resolution MS... but no HRMS data were report. Please specify how the HR-MS has been used and add this data (accurate mass, and fragmentation) to describe the isolated compounds.

ANSWER: The high-resolution MS spectra of compounds 8 and 9 were available in the supplementary material as part of the first version of the manuscript. Based on the suggestion of the reviewer, this data (HR-MS, and fragments) is now summarized in the main text of the new version of the manuscript in page 5, lines 185-192.

COMMENT: Did the authors coupled the UV detector with mass spectrometry analysis? Please clarify this point

ANSWER: The high-resolution MS spectra of compounds 8 and 9 were obtained by direct infusion MS, therefore we could not couple UV detection to mass spectrometry. Nonetheless, on-line UV spectra of isolated compounds, as well as of the standard compounds were obtained by HPLC and are now displayed in the figure 1 in the new version of the manuscript. See figure 1 at page 3.

Reviewer 2 Report

The manuscript is very clearly written and the results are appropriately presented. The only suggestion relates to the presentation of NMR spectra, only if possible and if the authors consider is appropriate to present them. 

Author Response

Reviewer 2

COMMENT: The manuscript is very clearly written and the results are appropriately presented. The only suggestion relates to the presentation of NMR spectra, only if possible and if the authors consider is appropriate to present them.

ANSWER: We acknowledge the comments made by the reviewer. As suggested, we are moving table 1 from the main text to the supplementary material. 

Reviewer 3 Report

The present manuscript entitled “Dalbergia ecastaphyllum (L.) Taub. and Symphonia globulifera L.f.:, the botanical sources of isoflavonoids and benzophenones in Brazilian red propolis” As currently presented and written, the manuscript does not do justice to the importance of the work that has been performed. I recommend reject it because the topic is well covered I just did bibliographic search and I have found several articles published in 2020 here some are relevant to the topic. 1. https://doi.org/10.1016/j.biori.2019.02.001. 2. https://doi.org/10.1016/j.micres.2018.05.003 3. Saito, É., Sacoda, P., Paviani, L. C., Paula, J. T., & Cabral, F. A. (2020). Conventional and supercritical extraction of phenolic compounds from Brazilian red and green propolis. Separation Science and Technology, 1-8.‏ 4. Júnior, F., de Carvalho, J. H., Valadas, L. A. R., Fonseca, S. G. D. C., Lobo, P. L. D., Calixto, L. H. M., ... & Rodrigues Neto, E. M. (2020). Clinical and Microbiological Evaluation of Brazilian Red Propolis Containing-Dentifrice in Orthodontic Patients: A Randomized Clinical Trial. Evidence-Based Complementary and Alternative Medicine, 2020.‏ 5. Martorano-Fernandes, L., Cavalcanti, Y. W., & de Almeida, L. D. F. D. (2020). Inhibitory effect of Brazilian red propolis on Candida biofilms developed on titanium surfaces. BMC Complementary Medicine and Therapies, 20(1), 1-9.‏ 6. Fasolo, D., Pippi, B., Meirelles, G., Zorzi, G., Fuentefria, A. M., von Poser, G., & Teixeira, H. F. (2020). Topical delivery of antifungal Brazilian red propolis benzophenones-rich extract by means of cationic lipid nanoemulsions optimized by means of Box-Behnken Design. Journal of Drug Delivery Science and Technology, 56, 101573.‏ 7. Yuan, M., Yuan, X. J., Pineda, M., Liang, Z. Y., He, J., Sun, S. W., ... & Li, K. P. (2020). A comparative study between Chinese propolis and Brazilian green propolis: metabolite profile and bioactivity. Food & Function.‏ 8. Tomazzoli, M. M., Zeggio, A. R. S., Pai Neto, R. D., Specht, L., Costa, C., Rocha, M., ... & Maraschin, M. (2020). Botanical source investigation and evaluation of the effect of seasonality on Brazilian propolis from Apis mellifera L. Scientia Agricola, 77(6).‏ 9. Costa Jr, A. G., Yoshida, N. C., Garcez, W. S., Perdomo, R. T., Matos, M. D. F. C., & Garcez, F. R. (2020). Metabolomics Approach Expands the Classification of Propolis Samples from Midwest Brazil. Journal of Natural Products, 83(2), 333-343.‏

Author Response

Reviewer 3

COMMENT: The present manuscript entitled “Dalbergia ecastaphyllum (L.) Taub. and Symphonia globulifera L.f.:, the botanical sources of isoflavonoids and benzophenones in Brazilian red propolis” As currently presented and written, the manuscript does not do justice to the importance of the work that has been performed. I recommend reject it because the topic is well covered I just did bibliographic search and I have found several articles published in 2020 here some are relevant to the topic.

ANSWER: We acknowledge the comments made by the reviewer. As observed by the reviewer, propolis research has gained increasing attention because of their promising biological properties. However, propolis universe is very big, since there are many types of propolis classified based on its physicochemical characteristics and botanical sources.

The “Brazilian Red propolis” is a type of propolis that has received attention because of its economic importance in Brazilian propolis market. Then, in this manuscript, submitted as a communication, we deal with a fundamental question: which botanical species is the botanical source of polyprenylated benzophenones present in “Brazilian Red propolis”? Our efforts to answer to this question were guided by the numerous articles in scientific media stating that Brazilian red propolis contains polyprenylated benzophenones, but no one described the botanical origin of these constituents (from which plant does the bees collect these constituents?).

Moreover, polyprenylated benzophenones constituents present in “Brazilian Red Propolis” have been identified as important contributors to the biological activity observed for red propolis extracts (e.g. guttiferone E, in the references 1 and 2 listed by reviewer). Despite of the biological importance of these constituents, its botanical source was not previously known. Therefore, we are reporting for the first time that Symphonia globulifera constitutes the botanical source of these constituents. This finding is certainly of great importance for researchers working on the chemistry of red propolis, but also for beekeepers, whose will be able to extent the bee pasture of Symphonia globulifera, and consequently standardize the quality of red propolis.

We are also reporting, for the first time that isolated compound oblongifolin  corresponds to oblongifolin  B and not to oblongifolin A, as reported in the literature from Brazilian red propolis.

We are submitting the new version of the manuscript, based on the comments of two other reviewers, as well for the consideration of the present reviewer.

COMMENT Regarding the cited publications by the reviewer:

  1. Santos et al. (2019) Brazilian red propolis extracts: study of chemical composition by ESI-MS/MS (ESI+) and cytotoxic profiles against colon cancer cell lines. https://doi.org/10.1016/j.biori.2019.02.001.

ANSWER: This article describes Dalbergia ecastaphyllum as the botanical source of isoflavonoids and other phenolic, but no data is described on the botanical source of benzophenones (Guttifenone E and xanthochymol).

  1. Ruffato et al. (2018) Brazilian red propolis: Chemical composition and antibacterial activity determined using bioguided fractionation. https://doi.org/10.1016/j.micres.2018.05.003 3.

ANSWER: Similarly to the previous article, this article describes Dalbergia ecastaphyllum as the botanical source of isoflavonoids and other phenolic, but no data is described on the botanical source of benzophenones (Guttifenone E and oblongifolin A).

  1. Saito, É., Sacoda, P., Paviani, L. C., Paula, J. T., & Cabral, F. A. (2020). Conventional and supercritical extraction of phenolic compounds from Brazilian red and green propolis. Separation Science and Technology, 1-8.‏ 4.

ANSWER: Article in the area of supercritical extraction. Similarly to the previous article, this article describes Dalbergia ecastaphyllum as the botanical source of isoflavonoids and other phenolic, but no data is described on the botanical source of benzophenones.

  1. Júnior, F., de Carvalho, J. H., Valadas, L. A. R., Fonseca, S. G. D. C., Lobo, P. L. D., Calixto, L. H. M., ... & Rodrigues Neto, E. M. (2020). Clinical and Microbiological Evaluation of Brazilian Red Propolis Containing-Dentifrice in Orthodontic Patients: A Randomized Clinical Trial. Evidence-Based Complementary and Alternative Medicine, 2020.‏ 5.

ANSWER: This article describes the production of a dentifrice containing Brazilian Red Propolis. This article describes Dalbergia ecastaphyllum as the botanical source of Brazilian Red Propolis.

  1. Martorano-Fernandes, L., Cavalcanti, Y. W., & de Almeida, L. D. F. D. (2020). Inhibitory effect of Brazilian red propolis on Candida biofilms developed on titanium surfaces. BMC Complementary Medicine and Therapies, 20(1), 1-9.‏ 6.

ANSWER: This article describes the inhibitory effect of Brazilian red propolis on Candida biofilms. No data is presented or discussed about the botanical source of Brazilian Red Propolis.

  1. Fasolo, D., Pippi, B., Meirelles, G., Zorzi, G., Fuentefria, A. M., von Poser, G., & Teixeira, H. F. (2020). Topical delivery of antifungal Brazilian red propolis benzophenones-rich extract by means of cationic lipid nanoemulsions optimized by means of Box-Behnken Design. Journal of Drug Delivery Science and Technology, 56, 101573.‏ 7.

ANSWER: This article describes the production of cationic lipid nanoemulsion. The authors use a benzophenone-rich extract, but no data is presented or discussed about the botanical source of this compounds on Brazilian Red Propolis.

  1. Yuan, M., Yuan, X. J., Pineda, M., Liang, Z. Y., He, J., Sun, S. W., ... & Li, K. P. (2020). A comparative study between Chinese propolis and Brazilian green propolis: metabolite profile and bioactivity. Food & Function.‏ 8.

ANSWER: Article deals with Brazilian Green Propolis, a kind of propolis produced in a different phytogeographic region of Brazil having Baccharis dracunculifolia as the botanical source.

  1. Tomazzoli, M. M., Zeggio, A. R. S., Pai Neto, R. D., Specht, L., Costa, C., Rocha, M., ... & Maraschin, M. (2020). Botanical source investigation and evaluation of the effect of seasonality on Brazilian propolis from Apis mellifera L. Scientia Agricola, 77(6).‏ 9.

ANSWER: This article deals with Brazilian Green Propolis, a kind o propolis produced in a different phytogeographic region of Brazil for which Baccharis dracunculifolia is the botanical source.

  1. Costa Jr, A. G., Yoshida, N. C., Garcez, W. S., Perdomo, R. T., Matos, M. D. F. C., & Garcez, F. R. (2020). Metabolomics Approach Expands the Classification of Propolis Samples from Midwest Brazil. Journal of Natural Products, 83(2), 333-343.‏

ANSWER: This article presents a metabolomics approach for classification purposes of propolis samples produced in Midwest Brazil, a different phytogeographic region of Brazil. The article reviews current knowledge on the chemistry and the botanical sources of propolis samples. Additionally, the article highlights Dalbergia ecastaphyllum as the botanical source of isoflavonoids in Brazilian Red Propolis, but no data is described on the botanical source of benzophenones, but only that some Clusia species (Clusiaceae) maybe suggested as sources of prenylated benzophenones.

Round 2

Reviewer 3 Report

The authors answered and clarified my questions regarding to the reported articles.